# Polypharmacotherapy in Psychiatry: Global Insights from a Rapid Online Survey of Psychiatrists

**DOI:** 10.3390/jcm11082129

**Published:** 2022-04-11

**Authors:** Michal Ordak, Daria Tkacz, Aniela Golub, Tadeusz Nasierowski, Magdalena Bujalska-Zadrozny

**Affiliations:** 1Department of Pharmacodynamics, Centre for Preclinical, Research and Technology (CePT), Medical University of Warsaw, 1B Banacha Street, 02-097 Warsaw, Poland; s080971@student.wum.edu.pl (D.T.); aniela.zabielska1@gmail.com (A.G.); mbujalska@wum.edu.pl (M.B.-Z.); 2Department of Psychiatry, Medical University of Warsaw, 1B Banacha Street, 02-097 Warsaw, Poland; tadeusz@wum.edu.pl

**Keywords:** polypharmacotherapy, hepatotoxicity, psychotropic drugs, drug interactions

## Abstract

In recent years, an increase in the problem of polypharmacotherapy in psychiatric patients has been observed, including the widespread problem of groups of people taking new psychoactive substances. One reason for this problem may be the poor knowledge of pharmacological interactions in psychiatry. The aim of this study was to explore the opinions and knowledge of psychiatrists from around the world on various aspects related to polypharmacotherapy. A total of 1335 psychiatrists from six continents were included in the study. The respondents’ opinion on the problem of hepatotoxicity in psychiatry was also examined. The greatest discrepancy among psychiatrists from different continents in the answers given concerned the definition of polypharmacotherapy (*p* < 0.001) and the approach to hepatotoxicity (*p* < 0.001). It is noteworthy that only about 20% of the psychiatrists surveyed (*p* < 0.001) believe that polypharmacotherapy is associated with a higher rate of patients’ hospitalisations. The most commonly used type of polypharmacy by psychiatrists was antidepressants and antipsychotics. Most of them also stated that polypharmacy was associated with reduced patient compliance with the doctor’s recommendations related to taking medications due to the increased complexity of the therapy. The continent that diversified the analysed questions to the greatest extent was Africa. Future educational activities for trainee psychiatrists should include more discussion of polypharmacotherapy in psychiatry.

## 1. Introduction

One of the major problems of modern pharmacotherapy in psychiatry is related to selecting the right combination of drugs for patients suffering from different conditions and requiring multidrug therapy. These patients constitute a group of people with a high risk of drug interactions. These interactions may result in a reduction of the therapeutic effect, the appearance of adverse reactions, and, in some situations, very seriously endanger the health and life of the patient. Post-drug complications increase the morbidity of the population, and, consequently, affect the costs borne by health systems, mainly due to an increase in the frequency of medical visits and hospitalisations. Polypharmacotherapy is commonly defined as the use of five or more medications daily by an individual, but is still debated, and it can vary from 2 to 11 concurrent medications. There is a diversity of definitions of polypharmacotherapy [1,2]. Polypharmacotherapy is particularly dangerous when the patient is treated by multiple specialists who are not necessarily informed which drugs the patient is currently taking [3].

Up to a third of outpatient psychiatric patients in the United States were receiving three or more psychotropic drugs at the same time. Over time, polypharmacotherapy has been used in an increasing proportion of patients [1]. According to data published in 2021, in a group of 14,418 hospitalised cases, 31% received at least five drugs simultaneously. Less than half of all patients received a combination of different antidepressants, or had antidepressant treatment supported by antipsychotics. The risk of any potential drug-drug interaction (pDDI) and potentially inappropriate medication (PIM) increased with each additional medication [4]. In a group of 47,071 inpatient hospital admissions, one-third of cases received an average of at least five different medications per day during their hospital stay. Fifty-one per cent of patients were receiving more than one psychotropic drug at a time [5]. According to data from 2021, the use of three or more psychotropic drugs in paediatrics increased significantly in the United States between 2000 and 2020. A higher number of side-effects were associated with drug regimens. Factors contributing to this include incorrect assumptions about the efficacy of combinations, as well as limited professional awareness of the metabolic and neurological side-effects of drugs [6]. The latest results of other studies carried out in a group of 448 psychiatric patients indicated that polypharmacotherapy was a risk factor for single and multiple drug-related adverse reactions [7].

The problem of polypharmacotherapy in patients taking various types of new psychoactive substances (NPS), such as mephedrone, has attracted attention in recent years. Ninety-nine per cent of the patients who were taking mephedrone combined it with other psychoactive substances, which resulted in the need for polypharmacotherapy. It increases the risk of subsequent hospitalisation of the same patients. This practice leads to a vicious circle effect resulting in numerous interactions not only between the drugs used, but also between abused psychoactive substances [8,9].

No data are observed in the literature regarding the knowledge of psychiatrists from all over the world on different aspects related to the topic of polypharmacotherapy. A single study, i.e., in a group of 347 psychiatrists from 34 countries, showed that their decision-making regarding polypharmacotherapy is policy-dependent. Another antipsychotic polytherapy survey conducted in a group of 44 prescribers showed that it was mainly assigned by those with more clinical experience and fewer concerns about it [10]. In Japan, in a group of 217 psychiatrists, the use of antipsychotic polypharmacotherapy was associated with, among other things, practice in a psychiatric hospital, as well as concern about potential drug–drug interactions [11]. In a group of Nigerian psychiatrists, polypharmacotherapy was mainly associated with the use of first-generation antipsychotics, as well as a desire to reduce the number of non-psychotic drugs [12].

The aim of our study was to investigate the attitudes of psychiatrists from six continents on various aspects related to polypharmacotherapy, including its definition, type of risks and benefits, and reasons for use. One of the main factors indicating the need for this type of study was the growing problem of polypharmacotherapy in patients taking new psychoactive substances in recent years. Studies conducted in the last 2 years have shown that only about 14% of psychiatrists know the names of and use pharmacological interaction databases in their daily work [13]. The attitudes towards the problem of hepatotoxicity associated with medications used in psychiatry were further investigated in the study group of psychiatrists. Polypharmacotherapy results in patients taking more and more medicinal products, which, by interacting pharmacokinetically and pharmacodynamically, induce symptoms of drug-related diseases, of which drug-related hepatopathies are one of the most clinically relevant [14]. An example of this would be patients taking a new psychoactive substance such as mephedrone, which is combined by most patients with other psychoactive substances, resulting, very often, in the need for polypharmacotherapy. The severity of the hepatotoxicity problem associated with the ingestion of psychoactive substances, combined with a range of potentially hepatotoxic drugs, can only make it more difficult to achieve a therapeutic effect [9]. For this reason, the authors decided to further investigate the opinion and knowledge of psychiatrists on hepatotoxicity in psychiatry.

## 2. Methods

### 2.1. Studied Group and Procedure

A total of 1335 psychiatrists from six continents were included in the analysis. Contact with psychiatrists was mainly through the LinkedIn Recruiter portal. Each person taking part in the study was made aware of the purpose of the survey before agreeing to complete it. Participation in the study was voluntary. The first questions concerned sociodemographic data, as well as daily medical practice. The next questions addressed several aspects related to the polypharmacotherapy used, i.e., its definition, reasons/types of use, possible ways of its reduction, factors influencing the occurrence, and risks/benefits associated with it (Appendix A). The questionnaire also included two questions on the appropriateness of polypharmacotherapy in schizophrenia and bipolar affective disorder. In line with the aim of the study outlined in the introduction, the question on hepatotoxicity in psychiatry, including the one associated with the use of psychotropic drugs, was also included in the study.

### 2.2. Statistical Analysis

In order to check whether there are statistically significant differences between psychiatrists from different continents in terms of the variable measured on the ordinal scale, the Kruskal–Wallis test was used. Selection was based on homogeneity of variance in the groups of individuals being compared. Effect size was measured by the epsilon-squared coefficient. For questions measured on a nominal scale, the chi-squared test and the Fi–Cramer coefficient were used. A *p* value < 0.05 was taken as the statistically significant level.

## 3. Results

### 3.1. Characteristics of the Studied Group of Psychiatrists

The study involved 1335 psychiatrists from six continents. The largest part of the study group was made up of psychiatrists of European origin, aged less than 40 years, with work experience ranging from 1 to 10 years, as well as men (Table 1).

### 3.2. Definition of Polypharmacotherapy

The definition of polypharmacotherapy was evaluated differently by the study group of psychiatrists, i.e., from different continents. This is indicated, among other things, by the result of Cramer’s V-factor. The largest number of psychiatrists stated that polypharmacotherapy involves a patient taking two or more drugs as part of the treatment of a disease or disorder for a period of time sufficient to overlap the effect of these drugs. The exception here are psychiatrists of African descent, who indicated as the most frequently selected answer the simultaneous use of two or more drugs by the patient as part of the treatment of a disease or disorder (Table 2).

### 3.3. Reasons for Using Polypharmacotherapy

A greater proportion of the psychiatrists surveyed believe that monotherapy for a given disease does not have the expected effect according to clinical trials. Most of them expressed an affirmative view that the high complexity of the disorder or its severe course makes it difficult to choose an appropriate monotherapy. There are no statistically significant differences between continents in terms of individual reasons (Table 3).

### 3.4. Type of Polypharmacotherapy

Among the types of polypharmacotherapy used by the surveyed group of psychiatrists, the most frequently chosen answer was the use of antidepressants with antipsychotics. Slightly more psychiatrists from Africa, compared to other continents, said they use two or more antipsychotic medications. In relation to the other continents, fewer respondents chose the answer of using antidepressants with sedative–sleep medication (Table 4).

### 3.5. Clozapine Pharmacotherapy in the Treatment of Schizophrenia

Among the statements concerning the pharmacotherapy with clozapine in the treatment of schizophrenia, the most frequently chosen answer concerned the opinion that the use of clozapine monotherapy may be associated with less frequent hospitalisation than in the case of antipsychotic polypharmacotherapy during the treatment of schizophrenia. The other statements listed in the table below (Table 5) were selected by a small percentage of the psychiatrists surveyed. Monitoring of liver enzyme levels during clozapine pharmacotherapy has been indicated mainly by psychiatrists from Australia and South America. The greatest variation in the answers given concerns the knowledge that clozapine monotherapy has no confirmed and described side-effects associated with hepatotoxicity.

Only 31.6% of the psychiatrists surveyed believe that while taking high-dose clozapine, liver enzyme levels should be monitored to avoid potential side-effects related to hepatotoxicity. This type of statement enters into a statistically significant relationship with the gender of psychiatrists from individual continents. The greater part of the surveyed women, compared to men, indicated that they agree with this type of aspect (Figure 1). This concerns North America (*p* < 0.001) and South America (*p* < 0.001).

### 3.6. Polypharmacotherapy in the Treatment of Bipolar Disorder

A majority of psychiatrists have stated that for depressive states in bipolar disorder, it is acceptable to combine two or even three drugs with different mechanisms of action or those that have synergistic effects. The remaining answers related to bipolar disorder were chosen by a small percentage of the surveyed group. A smaller percentage of psychiatrists from Europe compared to other continents stated that patients undergoing monotherapy in the treatment of bipolar disorder were more likely to have recurrent acute manic episodes than patients on polypharmacotherapy. The greatest variation in the answers given concerns the first two statements listed in the table below (Table 6).

One of the aspects mentioned in the above table enters into a statistically significant correlation in all continents. It concerns whether, in depressive states in bipolar disorder, it is permissible to combine two or even three drugs with different mechanisms of action or those that have a synergistic effect. In each of the individual continents, a greater percentage of the surveyed women chose this type of response as compared to men (*p* < 0.001, Figure 2).

### 3.7. Hepatotoxicity and the Effects of Psychiatric Drugs on Liver Function

A greater proportion of psychiatrists from across the continents said that knowledge of the influence of drugs on the functioning of the liver is useful in planning therapy in psychiatric patients taking more than one drug. This applies above all to European psychiatrists. More than 90% of psychiatrists chose the last aspects listed in Table 5 related to the hepatotoxicity of psychotropic drugs, although this applies to a slightly lesser extent to the subjects of African origin. Of note is the fact that a significant proportion of psychiatrists, mainly from Asia, believe that hepatotoxicity is well-studied, and described side-effect of many drugs used in psychiatric practice. The greatest variation in the study group of psychiatrists concerns the first and third aspects listed in the table below related to hepatotoxicity (Table 7).

### 3.8. Prevent and/or Reduce Excessive Polypharmacotherapy in Psychiatric Patients

Among the various methods associated with preventing and/or reducing excessive polypharmacotherapy among psychiatric patients, the psychiatrists surveyed from each continent indicated all the methods listed in Table 8 as effective (1–5 from ineffective to effective).

We also observed statistically significant differences between women and men in terms of the factor of cooperation with specialists in the field of pharmacology or pharmacists in order to review the list of drugs taken by patients with polypharmacy. This applies to continents such as Asia, Europe, and North and South America. In these continents, women have expressed a greater view that this factor may prevent and/or reduce excessive polypharmacy in psychiatric patients. In men from these continents, the median of the answers given was four, and in women it was higher, i.e., equal to five. In other words, women from these continents believe, to a greater extent, that the analysed factor may contribute to the prevention and/or reduction of excessive polypharmacy among psychiatric patients:−Asia, U = 4493; *p* < 0.001−Europe, U = 13,256; *p* < 0.001−North America, U = 4668.5; *p* < 0.001−South America, U = 11,697.5; *p* < 0.001

### 3.9. Possible Risks/Benefits Associated with the Use of Polypharmacotherapy

Among the possible risks/benefits associated with the use of polypharmacotherapy, the largest number of psychiatrists surveyed stated that polypharmacotherapy is associated with reduced patient compliance with the doctor’s recommendations related to taking medications due to the increase in the complexity of the therapy. This type of sentence was mainly indicated by psychiatrists coming from Europe. The same applies to their view that the introduction of a second drug can lead to a reduction in the occurrence of the negative side-effects of the first drug, e.g., weight gain during psychiatric treatment, while maintaining the therapeutic effect. The largest number of Australian-origin psychiatrists in relation to other psychiatrist groups believe that polypharmacotherapy may lead to prescribing too high doses of drugs. A small percentage of psychiatrists from each continent believe that polypharmacotherapy may be associated with higher rates of patient hospitalisation. The same applies to their view that, for some patients treated with antipsychotics, the therapeutic effect of polypharmacotherapy is better than that of monotherapy, particularly when analysing therapy with clozapine. The greatest variation in the answers given relates to the opinion that polypharmacotherapy is associated with a decrease in patient compliance with the doctor’s instructions related to taking medication due to an increase in the complexity of the therapy (Table 9).

It is noteworthy that in individual continents, the greater part of the surveyed women, compared to men, indicated the second, third, and fifth aspects related to polypharmacotherapy mentioned in the above table (Figure 3).

### 3.10. The Factors Influencing the Occurrence of the Phenomenon of Polypharmacotherapy

Among the factors influencing the occurrence of the phenomenon of polypharmacotherapy, the greatest number of psychiatrists indicated the presence of comorbidities, the severity of the disease, and resistance to treatment. This is particularly true of European psychiatrists. A small percentage of psychiatrists, mainly from Europe, identified demographic factors as the answer. Individuals indicated socioeconomic status. The divergence of responses for the other factors is noteworthy. Among the psychiatrists, 50–60% believe that personality disorders and substance abuse may be such factors. The factor that differentiated the groups of psychiatrists to the greatest extent was comorbidities (Table 10). 

One of the least frequently indicated factors, i.e., demographic ones, draws attention. They were indicated by a small percentage of psychiatrists, and, as it turns out, they enter into a statistically significant correlation with their gender in individual continents. It turns out that more women than men stated that these demographic factors influence the occurrence of polypharmacotherapy. The largest differences in terms of sex are characteristic of demographic factors in Europe (*p* < 0.001). More women than men on continents also indicated factors such as personality disorders and substance abuse. The biggest differences in this case are in North America (*p* < 0.001) (Figure 4).

## 4. Discussion

To our knowledge, the study we conducted is the first to address the views of psychiatrists from six continents on the broad topic of polypharmacotherapy. In the first step, we assessed its definition, which was interpreted in a different way. Polypharmacotherapy is defined as the taking of two or more drugs by a patient as part of the treatment of a disease or disorder for a period of time sufficient to overlap the effect of these drugs [15,16]. Unfortunately, this type of answer was indicated by less than half of the psychiatrists surveyed. The initial answers listed in the first question are not correct due to the fact that they do not take into account any timeframe related to polypharmacotherapy, i.e., they do not provide any reference to the health of patients in whom each of the drugs taken is necessary. Some drugs may only be taken, e.g., in the hospital setting intermittently (so the overlap effect may not be observed), and some have to be taken due to chronic diseases, and hence, drug interactions become a significant problem.

The next aspect analysed in the study group of psychiatrists concerned their opinion on the reasons for using polypharmacotherapy in psychiatric patients. As the psychiatric diagnosis is based on the medical history, and the medical history is based on the doctor’s experience, it is often possible that the polypharmacotherapy applied is based more on the personal treatment methods of the professional concerned than on EBM (evidence-based medicine). An example is the situation of relying on the assumption that certain drugs work better taken in combination because it has worked for another patient, rather than on making sure that it is the safe and best form of therapy for that patient. 

The same applies, for example, to the belief that a patient’s symptoms are so complex that a number of drugs should be used at once rather than starting with single drugs in low doses. This question also aimed to find out whether doctors refer more to the literature and research or to their own experience in medical practice when making a final decision about the type of therapy. The median of the answers given indicates that a larger proportion of psychiatrists believe that monotherapy in a given disease does not bring the expected results according to clinical trials. The highest agreement was for the sentence indicating that the high complexity of the disorder or its severe course makes it difficult to choose an appropriate monotherapy. 

The psychiatrists surveyed stated that the most common type of polypharmacotherapy they used was a combination of antidepressants and antipsychotics. The number of visits with two or more drugs prescribed increased from 42.6% in 1996–1997 to 59.8% in 2005–2006. For the number of visits with three or more drugs, the increase ranged from 16.9% to 33.2%. Combinations of antidepressants with sedative–sleep medications (23.1%), antipsychotics (12.9%), and other antidepressants (12.6%) were the first, second, and third most commonly prescribed psychotropic drug combinations in the years when this type of study was conducted [1].

The next two questions analysed concerned clozapine pharmacotherapy for schizophrenia and pharmacotherapy for bipolar affective disorder. The largest number of psychiatrists surveyed chose the correct answer, namely that the use of clozapine monotherapy may be associated with less frequent hospitalisation than in the case of using antipsychotic polypharmacotherapy during the treatment of schizophrenia. While comparing patients diagnosed with schizophrenia and receiving Medicaid, in a 12-month study, Velligan et al. noted that clozapine monotherapy was associated with lower rates of disease-related hospital visits. The Medicaid cost difference was 40–45% compared with patients receiving polypharmacotherapy, most commonly a combination of risperidone and quetiapine, or risperidone and ariprazole [17]. The other two correct answers listed in Table 4, i.e., third and fourth, were indicated by a small percentage of the psychiatrists surveyed. This is especially true since clozapine is a monotherapy drug that can cause serious liver damage. The studies conducted so far indicate that hepatotoxicity should be monitored when using clozapine [18]. For example, in a 30-year-old man, the level of liver enzymes during the 4-week monotherapy with clozapine increased significantly after 3 weeks, so as not to allow further use of the drug [19]. Some differences in therapeutic doses of antipsychotics are known to be due to ethnicity (i.e., allele frequencies of pharmacogenes associated to psychotropic medications) [20,21]. More than half of the psychiatrists surveyed also chose the correct answer, indicating that for depressive states in bipolar disorder, it is acceptable to combine two or even three drugs with different mechanisms of action or those that have a synergistic effect. It is acceptable to combine two or three drugs for the treatment of depressive states in ChAD, and these drugs may have the same or different mechanisms of action [22]. A small percentage of psychiatrists from individual continents stated that the addition of lamotrigine to the quetiapine used reverses the pro-inflammatory effect of quetiapine in microglia cells. Quetiapine has strong pro-inflammatory effects in microglia cells, but these can be reversed when quetiapine is used together with lamotrigine or with other drug combinations that contain lamotrigine [23].

In our study, we also examined the approach of the subjects to the problem of hepatotoxicity in psychiatry. The majority of the psychiatrists surveyed indicated four correct answers regarding the aspects studied. It is noteworthy, however, that a smaller percentage of psychiatrists from individual continents indicated one of these correct answers. It indicates that drug-related hepatotoxicity can be unequivocally established when symptoms associated with liver damage occur after starting the drug, disappear after stopping the drug, and reappear after subsequent administration of the same drug [24]. For example, antidepressants can cause varying degrees of liver damage, including fatal damage. They are most often idiopathic and dose-dependent. They can develop within a few days to up to six months after starting treatment. For this reason, among others, it is very important in psychiatry to monitor the level of liver enzymes, including in psychiatric patients taking more than one drug [25].

According to the psychiatrists surveyed, all of the factors listed in Table 6 are the main methods of reducing excessive polypharmacotherapy. A study by Mary Brunette et al. showed that providing clinical staff with educational courses and audits, combined with feedback on the accuracy of prescribed medications, reduced the phenomenon of polypharmacotherapy in psychiatry. According to the authors of the study, thanks to these practices, the phenomenon of antipsychotic polypharmacotherapy decreased from 13.1% to 10.9% [26]. Another study showed that patient-directed education is a very important tool in reducing excessive polypharmacotherapy. Patients (65–95 years) taking benzodiazepines were studied. Of the recipients who received adequate knowledge, 62% had a conversation with their doctor and/or pharmacist about stopping treatment with benzodiazepines. After 6 months, 27% of patients in this group had stopped using benzodiazepines, compared with 5% in the control group [27]. 

In other studies, in order to select the best possible pharmacological treatment for patients residing in nursing homes, an online platform was introduced which allows participants to access relevant information regardless of time and place, and facilitates medical records and information exchange between doctors, nurses, and pharmacists. Pharmacists in this study tested the validity of pharmacotherapy by analysing the drugs that were prescribed to patients. After applying the intervention proposed in the study, the improvement in the adequacy of the drugs used was most evident in the residents with an inadequate Medication Appropriateness Index (MAI) baseline. The authors of the study and articles conclude that the study they propose can realistically improve the appropriateness of treatment for nursing home patients [28,29]. In 2003, a programme was set up in New York to speed up the recovery of psychiatric patients by simplifying their treatment. The software that was developed allowed doctors to visualise the treatment history of their patients, as well as those of their colleagues. The system allowed doctors to make better decisions in treating patients, and thus, contributed to a reduction in antipsychotic polypharmacotherapy [30]. 

The penultimate topic analysed was connected with the risks and benefits of using polypharmacotherapy. Of the five correct answers, most psychiatrists indicated the fifth one. It indicates that polypharmacotherapy is associated with a reduction in patient compliance with the doctor’s recommendations related to taking medications due to the increase in the complexity of the therapy [16]. The remaining correct answers were chosen by a significantly smaller percentage of psychiatrists from individual continents. According to the literature, polypharmacotherapy may be a rational choice when monotherapy does not have the desired effect or when a drug is needed to reduce the side-effects of other substances used, but it is associated with a number of disadvantages and additional health risks and costs for patients, as well as the risk of burdening the health system [2,16,31]. For this reason, additional educational activities related to the aspects studied should be carried out in the future. This mainly concerns the fact that polypharmacotherapy may lead to prescribing too high doses of drugs, and may be associated with a higher percentage of patients hospitalisation.

One of the ways to reduce the problem of side-effects associated with drug interactions may be to involve pharmacists in educating psychiatrists on this subject [32]. Most of the data related to teaching psychopharmacology is contained in the American Society for Psychopharmacology Model Psychopharmacology Curriculum (ASCP). Educators present their suggestions related to teaching psychopharmacology [33]. Accordingly, one of the recommendations that could be made is to increase the emphasis in class on drug interactions in psychiatry. This applies, inter alia, to the use of databases in which you can check if there are side-effects associated with drug interactions. Another option for broadening the working knowledge of drug interactions among those educating psychiatry is by studying case reports, as well as reviewing the literature on psychotropic drugs [33,34,35].

Finally, we examined the aethiologic factors that may influence the occurrence of polypharmacotherapy. All of the factors listed in Table 7 play an important role here; however, two of them, i.e., socio-economic status and socio-demographics, were indicated by a smaller percentage of psychiatrists coming from each continent. It is also noteworthy that about 50% of psychiatrists identified personality disorders and substance abuse as factors. Lelliott et al. showed that in a group of patients aged 18–65 years, the risk of being prescribed more than one antipsychotic decreased by about 12% every ten years. The risk for men was about 1.4 times higher than for women [36]. 

The studies conducted by Haider et al. showed that people with the lowest level of education took the highest number of medications, and were most likely to experience polypharmacotherapy and excessive pharmacotherapy. The risk of polypharmacotherapy increased by an average of 11% as the level of education decreased. Women with low education had a slightly higher risk than men with low education. The researchers point out that this may be related to the fact that people with higher education have better access to knowledge related to their treatment and medication, and thus, communicate more actively with the specialist, and influence the prescribing process [37]. Kuno and Rothbard found that people of African American descent were less likely to receive antidepressants, but were more likely to receive higher doses of antipsychotics than Caucasians, despite similar severity of depressive symptoms. The overall conclusion of the researchers was that there are differences depending on the patient’s race that affect the treatment, perhaps due to the beliefs of doctors who expect lower compliance with medical recommendations by people of African American origin [38]. In a study by Lin and Smith, the authors note that black patients were more likely to be prescribed additional antipsychotics during treatment with olanzapine. Patients of African American descent were also more likely to receive higher doses of medication. The authors postulate that this is related to an overestimation of the risk of aggressive behaviour in these patients [39]. Burcu et al. found that, when considering visits to psychiatrists and non-psychiatrists, psychiatrists tended to prescribe antipsychotics more frequently to children and adolescents (24.2% vs. 4.6%). Rarely are these drugs prescribed as monotherapy, i.e., in only 10.2% of cases [40]. In their studies, Suzuki et al. found that some patients with drug-resistant schizophrenia (TRS) benefited from polypharmacotherapy with OLZ (olanzapine) and RIS (risperidone), which could not be achieved by monotherapy with either drug, albeit at the cost of side-effects, i.e., elevated blood prolactin levels, total cholesterol, and weight gain. They concluded that polypharmacotherapy may be beneficial for some patients difficult to treat, but not all, hence the need for further research and clarification of which patients may benefit from polypharmacotherapy and which may not [41]. In the elderly group (75 years and older), the number of medicines taken, and therefore, the frequency of polypharmacotherapy increases with the presence of comorbidities. Most of the subjects had five or more comorbidities, putting them at high risk of excessive polypharmacotherapy, with the greatest number of patients being prescribed antidepressants. The authors noted that there is a trend towards polypharmacotherapy in elderly patients with comorbidities such as hypertension, history of myocardial infarction, diabetes, heart failure, or dementia [42]. In monolingual Hispanics and African Americans, adherence to treatment recommendations was lower (77 and 68%) than in Caucasians (90%). The researchers additionally point out that diet affects the pharmacokinetic properties of drugs, and dietary habits vary among ethnic groups, which may affect the effects of drugs. In addition, ethnic minorities are often in the habit of using traditional herbal medicines, which may affect the effectiveness of the therapy [43]. As for the factor of other psychoactive substances, Barnes et al. point out that there is often a tendency to abuse them in patients with schizophrenia, which probably makes it difficult to adhere to prescribed pharmacotherapy. Patients who were additionally prescribed benzodiazepines, and who had a history of addiction and comorbidities, had a higher risk of benzodiazepine dependence and reported a lower quality of life standard [44,45].

## 5. Conclusions

Polypharmacotherapy might have bad consequences if being inappropriate, which should be further explored. Future education should place more emphasis on the topic of polypharmacotherapy in psychiatric patients. The diverse definitions of polypharmacotherapy, and the low awareness of its impact on the risk of subsequent hospitalizations of psychiatric patients, prove the need to strengthen the teaching of the medical community in this field. Attention should also be paid to the problem of hepatotoxicity in psychiatry, namely by making the medical community aware of the need to monitor laboratory parameters related to liver function. Further studies are required to assess to which extent the polypharmacy practice was rational and appropriate.

## 6. Limitations

A limitation of the study we conducted is remote contact with psychiatrists (it was an online survey), which has to do with the prevailing COVID-19 pandemic resulting in difficulties in doing inpatient work or attending psychiatric conferences. The sample usually consists of younger population when it is an online survey. However, research to date has shown that it is an effective portal for conducting this type of research. Another limitation of the study is the lack of the study conducted through other forms of contact with psychiatrists, such as through the website of the World Psychiatric Association (www.wpanet.org, accessed on 1 April 2022). Future research should consider an even larger group of psychiatrists, so as to be able to check whether there are statistically significant differences between women and men in terms of the examined aspect.

## Figures and Tables

**Figure 1 jcm-11-02129-f001:**
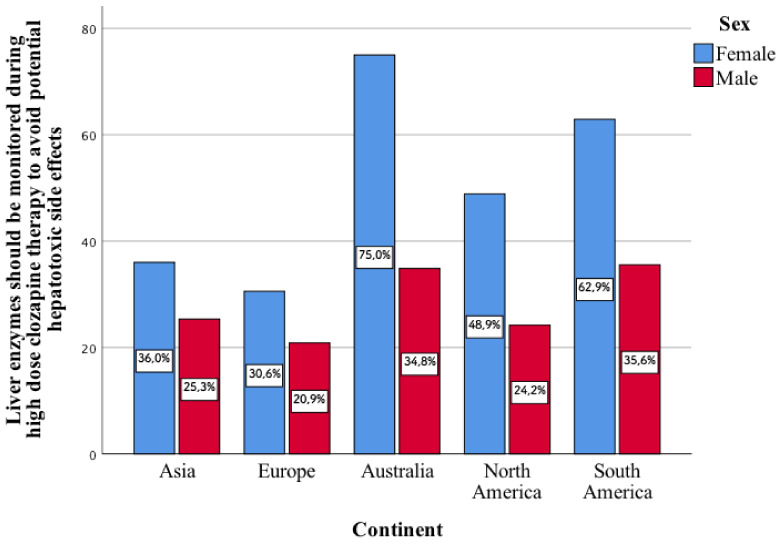
Opinion of the women and men surveyed on whether, in their opinion, the level of liver enzymes should be monitored during high-dose clozapine therapy in order to avoid potential side-effects related to hepatotoxicity.

**Figure 2 jcm-11-02129-f002:**
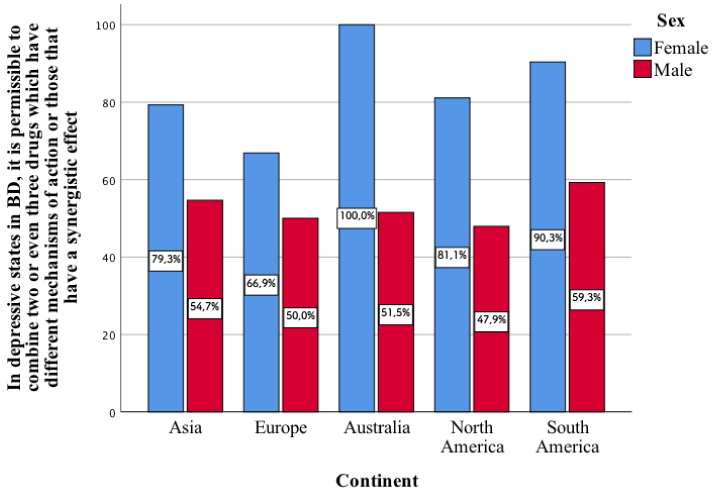
Opinion of the surveyed women and men on whether, in depressive states in bipolar disorder, it is permissible to combine two or even three drugs with different mechanisms of action or those that have a synergistic effect.

**Figure 3 jcm-11-02129-f003:**
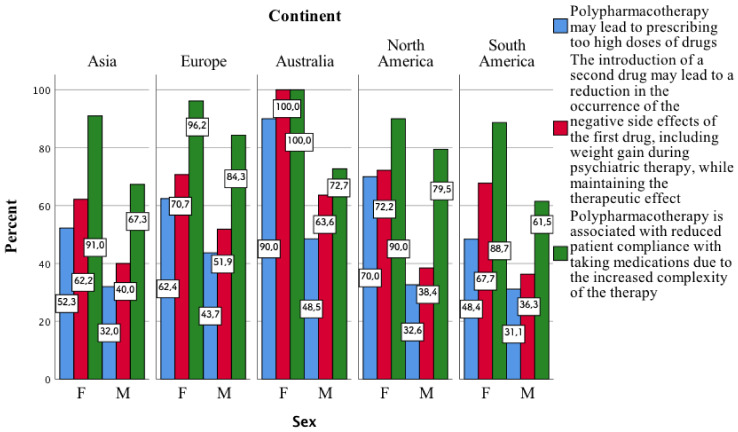
Opinion of the surveyed women and men on three aspects related to the risks and benefits of using polypharmacy.

**Figure 4 jcm-11-02129-f004:**
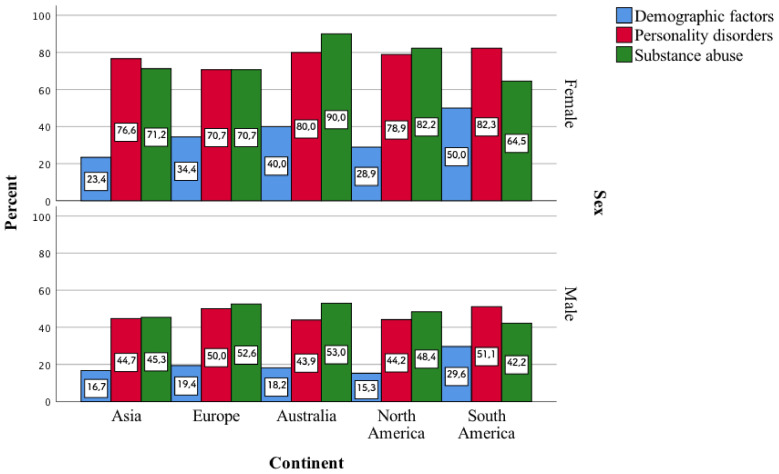
Demographic factors, personality disorders, and substance abuse as variables influencing, according to the surveyed women and men, the occurrence of the phenomenon of polypharmacotherapy.

**Table 1 jcm-11-02129-t001:** Sociodemographic data of the surveyed group of psychiatrists.

Variable		*n*	%	Statistical Test Result *
Continent	Asia	261	19.6	χ^2^(5) = 376.22; *p* < 0.001
	Europe	425	31.8	
	Australia	86	6.4	
	Africa	86	6.4	
	North America	280	21	
	South America	197	14.8	
Sex	Male	895	67	χ^2^(1) = 155.08; *p* < 0.001
	Female	440	33	
Age (years)	<40	774	58	χ^2^(2) = 629.16; *p* < 0.001
	41–60	523	39.2	
	61–80	38	2.8	
Seniority (years)	1–10	894	67	χ^2^(3) = 1343.13; *p* < 0.001
	11–20	269	20.1	
	21–30	147	11	
	>30	25	1.9	

* chi-squared test.

**Table 2 jcm-11-02129-t002:** Psychiatrists’ answers on the definition of polypharmacotherapy.

Definition of Polypharmacotherapy	Asia		Europe		Australia		Africa		North America		South America		Statistical Test Result *	Overall	
	*n*	%	*n*	%	*n*	%	*n*	%	*n*	%	*n*	%		*n*	%
A patient taking two or more drugs simultaneously to treat a disease or disorder	76	29.1	125	29.4	23	26.7	34	39.5	66	23.6	49	24.9	χ^2^(20) = 58.55; *p* < 0.001; V_cr_ = 0.21	373	27.9
A patient taking two or more drugs from the same category simultaneously to treat a disease or disorder	40	15.3	27	6.4	7	8.1	12	14	21	7.5	13	6.6		120	9
A patient taking two or more drugs from different categories simultaneously to treat a disease or disorder	39	14.9	71	16.7	16	18.6	10	11.6	43	15.4	26	13.2		205	15.4
A patient taking two or more drugs to treat a disease or disorder for a period sufficient for the effects of these drugs to overlap	73	28	163	38.4	28	32.6	23	26.7	103	36.8	95	48.2		485	36.3
A patient taking two or more drugs from any drug groups, including drugs unrelated to the therapy of the disease or disorder (such as NSAIDs)	33	12.6	39	9.2	12	14	7	8.1	47	16.8	14	7.1		152	11.4

* chi-squared test.

**Table 3 jcm-11-02129-t003:** Reasons for using polypharmacotherapy by the surveyed group of psychiatrists.

Reasons for Using Polypharmacotherapy (5—I Completely Agree, 3—It Is Difficult to Say, 1—I Do Not Agree at All)	Median						Statistical Test Result *	Overall
	Asia	Europe	Australia	Africa	North America	South America		
Monotherapy for a given disease does not bring the expected results according to clinical trials	2	3	2	2	2	2	H = 5.17; *p* = 0.4	2
Monotherapy for a given disease does not bring the expected results in the history of a given patient	2	2	2	2	2	1	H = 4.66; *p* = 0.46	2
Polypharmacotherapy was successful in other patients with similar symptoms or disorders	2	3	2	2	3	2	H = 3.13; *p* = 0.68	2
The high complexity of the disorder or its severe course makes it difficult to select an appropriate monotherapy	4	4	3	3	4	4	H = 6.03; *p* = 0.3	4
The use of combined drugs gives a better therapeutic effect than the use of one drug in other patients with a similar disorder/symptoms.	2	2	1	1	2	1	H = 10.29; *p* = 0.07	2

* Kruskal–Wallis test.

**Table 4 jcm-11-02129-t004:** Types of polypharmacotherapy used by the surveyed group of psychiatrists.

Type of Polypharmacotherapy	Asia		Europe		Australia		Africa		North America		South America		Statistical Test Result *	Overall	
	*n*	%	*n*	%	*n*	%	*n*	%	*n*	%	*n*	%		*n*	%
Two or more antidepressant drugs	78	29.9	137	32.2	22	25.6	23	26.7	78	27.9	52	26.4	χ^2^(5) = 3,74; *p* = 0.59; V_cr_ = 0.05	390	29.2
Two or more antipsychotic drugs	82	31.4	152	35.8	32	37.2	45	52.3	102	36.4	68	34.5	χ^2^(5) = 12.6; *p* = 0.03; V_cr_ = 0.1	481	36
Antidepressants with sedative–hypnotic drugs	127	48.7	199	46.8	37	43	31	36	124	44.3	113	57.4	χ^2^(5) = 14.25; *p* = 0.01; V_cr_ = 0.1	631	47.3
Antidepressants with antipsychotics	171	65.5	302	71.1	65	75.6	54	62.8	200	71.4	123	62.4	χ^2^(5) = 10.14; *p* = 0.07; V_cr_ = 0.09	915	68.5
Antipsychotics with sedative–hypnotic drugs	48	18.4	104	24.5	10	11.6	16	18.6	58	20.7	48	24.4	χ^2^(5) = 10.21; *p* = 0.07; V_cr_ = 0.09	284	21.3

* chi-squared test.

**Table 5 jcm-11-02129-t005:** Psychiatrists’ answers on the clozapine pharmacotherapy in the treatment of schizophrenia.

Clozapine Pharmacotherapy in the Treatment of Schizophrenia	Asia		Europe		Australia		Africa		North America		South America		Statistical Test Result *	Overall	
	*n*	%	*n*	%	*n*	%	*n*	%	*n*	%	*n*	%		*n*	%
The use of clozapine monotherapy may be associated with a reduced need for hospitalisation than in the case of antipsychotic polypharmacotherapy in the treatment of schizophrenia.	202	77.4	327	76.9	74	86	65	75.6	207	73.9	153	77.7	χ^2^(5) = 5.64; *p* = 0.34; V_cr_ = 0.07	1028	77
Polypharmacotherapy with sulpiride and clozapine does not show increased efficacy compared to clozapine monotherapy in the treatment of schizophrenia in patients who are refractory to treatment with atypical antipsychotics.	23	8.8	65	15.3	5	5.8	21	24.4	46	16.4	33	16.8	χ^2^(5) = 20.78; *p* = 0.001; V_cr_ = 0.13	193	14.5
Clozapine is a drug which can cause severe liver damage when used in monotherapy.	7	2.7	24	5.6	12	14	8	9.3	22	7.9	24	12.2	χ^2^(5) = 23.24; *p* < 0.001; V_cr_ = 0.13	97	7.3
Liver enzymes should be monitored during high dose clozapine therapy to avoid potential hepatotoxic side-effects.	78	29.9	104	24.5	38	44.2	25	29.1	90	32.1	87	44.2	χ^2^(5) = 31.32; *p* < 0.001; V_cr_ = 0.15	422	31.6
Clozapine monotherapy has no confirmed and described hepatotoxic side-effects.	46	17.6	69	16.2	25	29.1	8	9.3	33	11.8	54	27.4	χ^2^(5) = 32.03; *p* < 0.001; V_cr_ = 0.16	235	17.6

* chi-squared test.

**Table 6 jcm-11-02129-t006:** Psychiatrists’ answers on the polypharmacotherapy in the treatment of bipolar disorder.

Polypharmacotherapy in the Treatment of Bipolar Disorder	Asia		Europe		Australia		Africa		North America		South America		Statistical Test Result *	Overall	
	*n*	%	*n*	%	*n*	%	*n*	%	*n*	%	*n*	%		*n*	%
In the treatment of depression symptoms in BD, the addition of lamotrigine to quetiapine does not increase the efficacy of the pharmacotherapy.	8	3.1	12	2.8	7	8.1	11	12.8	17	6.1	25	12.7	χ^2^(5) = 35; *p* < 0.001; V_cr_ = 0.16	80	6
The addition of lamotrigine when using quetiapine reverses the pro-inflammatory effects of quetiapine in microglia cells.	6	2.3	6	1.4	5	5.8	7	8.1	11	3.9	21	10.7	χ^2^(5) = 35; *p* = 0.001; V_cr_ = 0.16	56	4.2
Patients undergoing monotherapy for the treatment of BD more often have relapses of acute manic episodes than patients undergoing polypharmacotherapy.	57	21.8	75	11.6	18	20.9	21	24.4	74	26.4	56	28.4	χ^2^(5) = 12.53; *p* = 0.03; V_cr_ = 0.1	301	22.5
Women with BD during pregnancy and in the puerperium period should undergo prophylactic polypharmacotherapy in order to avoid episodes of the disease during this period.	28	10.7	50	11.8	10	11.6	13	15.1	43	15.4	22	11.2	χ^2^(5) = 3.98; *p* = 0.55; V_cr_ = 0.06	166	12.4
In depressive states in BD, it is permissible to combine two or even three drugs which have different mechanisms of action or those that have a synergistic effect.	170	65.1	239	56.2	54	62.8	65	75.6	164	58.6	136	69	χ^2^(5) = 19.38; *p* = 0.002; V_cr_ = 0.12	136	69

* chi-squared test.

**Table 7 jcm-11-02129-t007:** Psychiatrists’ answers on the hepatotoxicity and the effects of psychiatric drugs on liver function.

Hepatotoxicity and the Effects of Psychiatric Drugs on Liver Function	Asia		Europe		Australia		Africa		North America		South America		Statistical Test Result *	Overall	
	*n*	%	*n*	%	*n*	%	*n*	%	*n*	%	*n*	%		*n*	%
Knowledge of the effects of drugs on liver function is useful when planning therapy for psychiatric patients taking more than one drug.	219	83.9	411	96.7	77	89.5	69	80.2	244	87.1	184	93.4	χ^2^(5) = 46.94; *p* < 0.001; V_cr_ = 0.19	1204	90.2
Hepatotoxicity is a well-studied and reported side-effect of many drugs used in psychiatric practice.	119	45.6	139	32.7	17	19.8	8	9.3	83	29.6	50	25.4	χ^2^(5) = 53.55; *p* < 0.001; V_cr_ = 0.2	416	31.2
Drug-related hepatotoxicity can be unequivocally determined when the symptoms of liver damage occur after a patient starts taking a drug, and disappear when the drug is discontinued.	22	8.4	49	11.5	13	15.1	21	24.4	13	4.6	40	20.3	χ^2^(5) = 44.3; *p* < 0.001; V_cr_ = 0.18	158	11.8
Drug-related hepatotoxicity can be unequivocally determined when the symptoms of liver damage occur after a patient starts taking a drug, disappear when the drug is discontinued, and reappear after the next administration of the same drug.	187	71.6	267	62.8	50	58.1	59	68.6	173	61.8	129	65.5	χ^2^(5) = 9.47; *p* = 0.09; V_cr_ = 0.08	865	64.8
When prescribing drugs, in particular, potentially hepatotoxic drugs, the patient should be required to test and monitor the level of liver enzymes.	250	95.8	404	95.1	82	95.3	73	84.9	267	95.4	189	95.9	χ^2^(5) = 18.32; *p* = 0.003; V_cr_ = 0.12	1265	94.8
Minimising polypharmacotherapy, in particular, with potentially hepatotoxic drugs, may lower the risk of liver damage in psychiatric patients.	254	97.3	411	96.7	82	95.3	77	89.5	273	97.5	190	96.4	χ^2^(5) = 13.7; *p* = 0.02; V_cr_ = 0.1	1287	96.4

* chi-squared test.

**Table 8 jcm-11-02129-t008:** Psychiatrists’ answers on preventing and/or reducing excessive polypharmacotherapy in psychiatric patients.

Effectiveness of the Use the Following Methods to Prevent And/or Reduce Excessive Polypharmacotherapy in Psychiatric Patients (1–5 from Ineffective to Effective)	Median						Statistical Test Result *	Overall
	Asia	Europe	Australia	Africa	North America	South America		
Education and training of hospital staff (doctors, nurses)	4	4	4	4	4	4	H = 8.85; *p* = 0.12	4
Conducting internal audits and controls	4	4	4	4	4	4	H = 4.45; *p* = 0.49	4
Educating patients and improving their understanding of the disease, the drug used, and its side-effects	4	4	4	4	4	4	H = 3.05; *p* = 0.69	4
Collaborating with specialists in the field of pharmacology or with pharmacists to review the list of drugs taken by patients in polypharmacotherapy	5	4	4	4	4	5	H = 7.06; *p* = 0.22	4
Introducing new programmes or applications into medical practice to help doctors show the medications taken previously by patients and their effectiveness.	4	4	4	4	5	4	H = 5.93; *p* = 0.31	4

* Kruskal–Wallis test.

**Table 9 jcm-11-02129-t009:** Psychiatrists’ answers on the risk/benefits of polypharmacotherapy.

Risk/Benefits of Polypharmacotherapy	Asia		Europe		Australia		Africa		North America		South America		Statistical Test Result *	Overall	
	*n*	%	*n*	%	*n*	%	*n*	%	*n*	%	*n*	%		*n*	%
For some patients treated with antipsychotics, the therapeutic effect of polypharmacotherapy is better than in the case of monotherapy, in particular, for clozapine therapy.	52	19.9	79	18.6	14	16.3	18	20.9	56	20	30	15.2	χ^2^(5) = 2.75; *p* = 0.74; V_cr_ = 0.05	249	18.7
Polypharmacotherapy may lead to prescribing too high doses of drugs.	106	40.6	215	50.6	50	58.1	40	46.5	125	44.6	72	36.5	χ^2^(5) = 18.97; *p* = 0.002; V_cr_ = 0.12	608	45.5
The introduction of a second drug may lead to a reduction in the occurrence of the negative side-effects of the first drug, including weight gain during psychiatric therapy, while maintaining the therapeutic effect.	129	49.4	250	58.8	62	72.1	55	64	138	49.3	91	46.2	χ^2^(5) = 28.26; *p* < 0.001; V_cr_ = 0.15	725	54.3
By ensuring faster and better therapeutic effects, polypharmacotherapy leads to better patient compliance with the doctor’s recommendations.	26	10	39	9.2	6	7	9	10.5	31	11.1	40	20.3	χ^2^(5) = 19.78; *p* = 0.001; V_cr_ = 0.12	151	11.3
Polypharmacotherapy is associated with reduced patient compliance with taking medications due to the increased complexity of the therapy.	202	77.4	377	88.7	68	79.1	69	80.2	232	82.9	138	70.1	χ^2^(5) = 35.21; *p* < 0.001; V_cr_ = 0.16	1086	81.3
Thanks to better therapeutic effectiveness, polypharmacotherapy leads to lower therapy costs and less frequent medical consultations/hospitalisation.	10	3.8	10	2.4	5	5.8	9	10.5	16	5.7	18	9.1	χ^2^(5) = 19.58; *p* = 0.002; V_cr_ = 0.12	68	5.1
Polypharmacotherapy may be associated with a higher rate of hospitalisation.	42	16.1	72	16.9	18	20.9	20	23.3	47	16.8	34	17.3	χ^2^(5) = 3.24; *p* = 0.66; V_cr_ = 0.05	233	17.5

* chi-squared test.

**Table 10 jcm-11-02129-t010:** Psychiatrists’ answers on the factors influencing the occurrence of the phenomenon of polypharmacotherapy.

Factor	Asia		Europe		Australia		Africa		North America		South America		Statistical Test Result *	Overall	
	*n*	%	*n*	%	*n*	%	*n*	%	*n*	%	*n*	%		*n*	%
Demographic factors (age, gender, race, patient’s level of education)	51	19.5	106	24.9	20	23.3	23	26.7	55	19.6	71	36	χ^2^(5) = 21.62; *p* = 0.001; V_cr_ = 0.13	326	24.4
Personality disorders	152	58.2	245	57.6	45	52.3	47	54.7	155	55.4	120	60.9	χ^2^(5) = 2.71; *p* = 0.75; V_cr_ = 0.05	764	57.2
The presence of comorbidities	208	79.7	385	90.6	68	79.1	70	81.4	249	88.9	182	92.4	χ^2^(5) = 30.38; *p* < 0.001; V_cr_ = 0.15	1162	87
The severity of the disease	230	88.1	397	93.4	77	89.5	73	84.9	259	92.5	185	93.9	χ^2^(5) = 12.87; *p* = 0.03; V_cr_ = 0.1	1221	91.5
Resistance to treatment	253	96.9	412	96.9	82	95.3	81	94.2	266	95	288	95.4	χ^2^(5) = 3.32; *p* = 0.65; V_cr_ = 0.05	1282	96
Socio-economic status	24	9.2	24	5.6	7	8.1	11	12.8	27	9.6	19	9.6	χ^2^(5) = 7.53; *p* = 0.18; V_cr_ = 0.08	112	8.4
Substance abuse	147	56.3	252	59.3	53	61.6	42	48.8	166	59.3	97	49.2	χ^2^(5) = 9.43; *p* = 0.09; V_cr_ = 0.08	757	56.7

* chi-squared test.

## Data Availability

Please contact the corresponding author with any data availability requests.

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
