# Peer review of "Polypharmacotherapy in Psychiatry: Global Insights from a Rapid Online Survey of Psychiatrists"

_jcm, 2022, doi:10.3390/jcm11082129_

Round 1

Reviewer 1 Report

The present manuscript is based on the important topic of polypharmacy (in psychiatry) and since it included a large sample of psychiatrists worldwide, the results of this study could be scientifically significant. Overall, this is well designed study, the methods are appropriate and the data are generally well described. However, there are some concerns that need to be clarified before publication.

Title

The authors should be consistent in the use of the terms ‘polypharmacy’ and ‘polypharmacotherapy’ throughout the text.

I suggest to rephrase the title, i.e. to exclude the part from 6 continents. In my opinion, ‘global insights’ indicates enough, and the details should be explained further in the manuscript.

Abstract

The first sentence (‘Studies conducted in recent years in patients abusing new psychoactive substances have shown a growing problem of polypharmacy’) implies that this issue exists only among the users of psychoactive substances, and it should be emphasized as a common issue in psychiatry in general.

The abstract generally should be more focused – more on results and less on methods (such as collecting data through LinkedIn).

Line 38 – ‘knowledge’ should be added besides the ‘opinions’ in the aim of the study.

Line 44 – what kind of discrepancy? among psychiatrists from different continents?

More key words should be added.

Introduction

There is no need for stating the names of the journals (throughout the whole manuscript), there are references and those who are interested in something specific may easily find all the details on the referred studies.

Lines 99-100: it should be clarified that 99% of the patients who were taking mephedrone combined it with other psychoactive substances (and not that almost all psychiatry patients were taking mephedrone).

Lines 111-113: Please add the reference.

Line 130: ‘knowledge’ should be added since it was also examined (not only opinion)

Methods

The manuscript would benefit from including the whole questionnaire as the supplement in order to follow the text easier.

Line 148: which ‘corresponding’ post-hoc test were used?

Results

Generally, the overall (global) results are missing in this section (for example, the ‘total’ column where appropriate). It should be at least mentioned in the text (if not in the table), what was the overall percentage of psychiatrists worldwide who gave some specific answer.

Another issue is the lack of official ‘polypharmacy’ definition. Polypharmacy is commonly defined as the use of five or more medications daily by an individual, but is still debated and it can vary from 2 to 11 concurrent medications. It should be added in the Introduction or explained more in-depth in Discussion section.

Line 165: ‘surveyed group of psychiatrists’ instead of ‘examined persons’

Lines 212-214: it is knowledge and not opinion.

Discussion

Again the issue with the definition of ‘polypharmacy’. The authors chose the definition from “articles published in BMC Geriatrics and BMC Psychiatry”. However, this diversity of definitions should be mentioned here.

Line 328: define EBM.

There are also unnecessary names of journals stated in the text of ‘Discussion’.

Lines 414-415 are very important and it should be mentioned in ‘Conclusion’ as well, that further studies are required to assess to which extent the polypharmacy practice was rational and appropriate.

Line 450: OLZ and RIS should be defined.

Conclusion

The ‘Conclusion’ should contain the summarized main results at least in one sentence. Besides, it should be emphasized that the ‘polypharmacy’ might have bad consequences if being inappropriate, which should be further explored.

Limitations of the study

There are more limitations that should be added. One of them is that it was online survey. The sample usually consists of younger population when it is an online survey.

Author Response

Dear Reviewer nr 1,

1. “The present manuscript is based on the important topic of polypharmacy (in psychiatry) and since it included a large sample of psychiatrists worldwide, the results of this study could be scientifically significant. Overall, this is well designed study, the methods are appropriate and the data are generally well described. However, there are some concerns that need to be clarified before publication.”

ANSWER: Thank you very much for receiving a positive review of the submitted manuscript. The responses to the comments received from the reviewer are provided below.

2. “I suggest to rephrase the title, i.e. to exclude the part from 6 continents. In my opinion, ‘global insights’ indicates enough, and the details should be explained further in the manuscript.”

ANSWER:Following the advice provided, the title of the manuscript has been edited.

3. „The first sentence (‘Studies conducted in recent years in patients abusing new psychoactive substances have shown a growing problem of polypharmacy’) implies that this issue exists only among the users of psychoactive substances, and it should be emphasized as a common issue in psychiatry in general.”

ANSWER: Following this type of clue, as well as identical advice from a third reviewer, the sentence has been edited. The edit was that this group of patients was included as an example (a frequent problem).

4. „The abstract generally should be more focused – more on results and less on methods (such as collecting data through LinkedIn).”

ANSWER:One sentence concerning the methodology has been removed from the abstract, i.e., related to the use of the LincedIn Recruiter portal. One sentence concerning the analysed aspects was also deleted, i.e., in order to include an additional sentence concerning the obtained results.

5. „Line 38 – ‘knowledge’ should be added besides the ‘opinions’ in the aim of the study.”

ANSWER:According to the given tip,the word was insertedwhat is knowledge for the purpose of research.

6. „Line 44 – what kind of discrepancy? among psychiatrists from different continents?”

ANSWER:The sentence was edited as indicated.

7. „More key words should be added.”

ANSWER:More keywords have been added.

8. „The authors should be consistent in the use of the terms ‘polypharmacy’ and ‘polypharmacotherapy’ throughout the text.”

ANSWER:As suggested next, the text uses one phrase, which is polypharmacotherapy.

9. „There is no need for stating the names of the journals (throughout the whole manuscript), there are references and those who are interested in something specific may easily find all the details on the referred studies.”

ANSWER: The names of journals have been removed from the sentences. The same is true for single sentences containing the name of the journal.

10. „Lines 99-100: it should be clarified that 99% of the patients who were taking mephedrone combined it with other psychoactive substances (and not that almost all psychiatry patients were taking mephedrone).”

ANSWER: The sentence was edited as indicated, i.e., the indicated specific percentage of patients was given.

11. „Lines 111-113: Please add the reference.”

ANSWER:Thank you for another correct suggestion, we have included the appropriate reference in the text.

12. „Line 130: ‘knowledge’ should be added since it was also examined (not only opinion)”

ANSWER: We added the word in the indicated place.

13. „The manuscript would benefit from including the whole questionnaire as the supplement in order to follow the text easier.”

ANSWER:The used questionnaire is included at the end of the manuscript.

14. „Line 148: which ‘corresponding’ post-hoc test were used?”

ANSWER:Thank you for this note. Our analyses using the Kruskal-Wallis test did not show any statistically significant differences, so the post-hoc test did not have to be applied. This sentence was written automatically (tradition when writing manuscripts), which in this case is obviously a mistake, because as we write, there were no statistically significant differences for this test. For this reason, this sentence has been removed.

15. „Generally, the overall (global) results are missing in this section (for example, the ‘total’ column where appropriate). It should be at least mentioned in the text (if not in the table), what was the overall percentage of psychiatrists worldwide who gave some specific answer.”

ANSWER:As suggested by the reviewer, a "total" column has been added to all tables. An additional total column was created for each table and the appropriately calculated data was placed.

16. „Another issue is the lack of official ‘polypharmacy’ definition. Polypharmacy is commonly defined as the use of five or more medications daily by an individual, but is still debated and it can vary from 2 to 11 concurrent medications. It should be added in the Introduction or explained more in-depth in Discussion section.”

ANSWER:In line with the suggestion obtained, this sentence has been included in the introduction.

17. „Line 165: ‘surveyed group of psychiatrists’ instead of ‘examined persons’

ANSWER:The signature of the indicated table was corrected.

18. „Lines 212-214: it is knowledge and not opinion.”

ANSWER:The word knowledge is placed in the indicated place.

19. „Again the issue with the definition of ‘polypharmacy’. The authors chose the definition from “articles published in BMC Geriatrics and BMC Psychiatry”. However, this diversity of definitions should be mentioned here.”

ANSWER:In line with the previous comment, the introduction includes an indication of the sentence concerning the definition of polypharmacotherapy. Additionally, in line with another suggestion, one reference to the diversity of the definitions of polypharmacotherapy was also added in the introduction.

20. „Line 328: define EBM”

ANSWER:The indicated abbreviation has been expanded.

21. „There are also unnecessary names of journals stated in the text of ‘Discussion’.”

ANSWER: As in the introduction, sentences were edited in the discussion by removing the names of journals from them.

22. „Lines 414-415 are very important and it should be mentioned in ‘Conclusion’ as well, that further studies are required to assess to which extent the polypharmacy practice was rational and appropriate.”

ANSWER:The indicated suggestion was included in the conclusions.

23. „Line 450: OLZ and RIS should be defined.”

ANSWER:As suggested by several reviewers, the abbreviations indicated have been expanded.

24. „The ‘Conclusion’ should contain the summarized main results at least in one sentence. Besides, it should be emphasized that the ‘polypharmacy’ might have bad consequences if being inappropriate, which should be further explored.”

ANSWER: The conclusions have been expanded to include the advice above and new guidance.

25. „There are more limitations that should be added. One of them is that it was online survey. The sample usually consists of younger population when it is an online survey.”

ANSWER:The article limitations have been extended.

Reviewer 2 Report

This is a rather descriptive study of essentially presenting questionnaire answers with limited interpretation value. For instance, it would help to compare the views of male and female psychiatrists, of those coming from different continents, addressing additional training of the psychiatrist etc. 

Author Response

Dear Reviewer nr 2,

„This is a rather descriptive study of essentially presenting questionnaire answers with limited interpretation value. For instance, it would help to compare the views of male and female psychiatrists, of those coming from different continents, addressing additional training of the psychiatrist etc.”

ANSWER: Thank you for your valuable feedback. In order to increase the transparency and quality of the manuscript, we have made a number of changes and corrections to the manuscript we sent. Among other things, as suggested by other reviewers: a total column was added to the tables. The abstract, conclusions and limitations have been edited. The discussion presents additional ways to improve the education of the future medical community about drug interactions and related side effects in the group of psychiatric patients. Based on other reviews obtained, the manuscript has been revised for many different small revisions. The aim of the study was to examine the opinions / knowledge of psychiatrists from 6 continents separately. However, with the valuable suggestion of the reviewer, we conducted additional analyses regarding the gender of the psychiatrists under study and described the statistically significant differences that were observed. They are presented in the form of a description and, in some cases, in the form of a graph. In the limitations of the manuscript, it is mentioned that the study should be conducted on a larger number of psychiatrists from different continents, i.e., so that comparisons can be made between women and men.

Reviewer 3 Report

JCM 1631425 Polypharmacotherapy in psychiatry: global insights from a rapid online survey of psychiatrists from 6 continents

This is an interesting work based on an online survey to psychiatrists around the world regarding their attitudes on polypharmacotherapy and pharmacological interactions in Psychiatry, which is growing in recent ears. The results of this study showed a great discrepancy in the definition of polypharmacotherapy and the approach to manage hepatotoxicity. It is urgent to include more discussion of polypharmacy in educational activities for psychiatrists.

-Why did the authors choose LinkedIn to perform the survey? Would not be better an “ad hoc” portal to psychiatrists (e.g., www.wpanet.org)?

-The term “psychoactive substances” was included since the abstract. However, the survey did not consider any question about these drugs and potential pharmacological interactions with psychotropic medications. Questions were more related to schizophrenia and bipolar affective disorder treatments (assuming as dual diagnoses?).

(e.g., “Studies conducted in recent years in patients abusing new psychoactive substances have shown a growing problem of polypharmacy).

-Authors should mention that some differences in therapeutic doses of antipsychotics (e.g. for clozapine in doi: 10.1055/a-1625-6388; DOI: 10.1016/j.euroneuro.2022.03.001) are known to be due to ethnicity (i.e. allele frequencies of pharmacogenes associated to psychotropic medications).

Minor comments:

-please, correct some misspellings:

Line 135 was mainly through the “Lincedin” Recruiter portal.

-The title of all tables appear after the tables are shown.

-In line 450, please, define abbreviations OLZ, and RIS.

Author Response

Dear Reviewer nr 3,

1. “This is an interesting work based on an online survey to psychiatrists around the world regarding their attitudes on polypharmacotherapy and pharmacological interactions in Psychiatry, which is growing in recent ears. The results of this study showed a great discrepancy in the definition of polypharmacotherapy and the approach to manage hepatotoxicity.”

ANSWER:Thank you very much for receiving a positive review of the submitted manuscript.The most important thing for us is the implementation of the obtained results, therefore we intend to present such results at the International Psychiatric Congress organized by the WPA. As suggested by another reviewer, in order to improve the quality of the manuscript, we also described statistically significant differences between women and men that were observed in this article.

2. „-Why did the authors choose LinkedIn to perform the survey? Would not be better an “ad hoc” portal to psychiatrists (e.g., www.wpanet.org)?”

ANSWER:The choice of this type of portal was related to the fact that it is an effective tool for acquiring contacts with a selected group of people. A specialized option of the LinkedIn portal, which is LinkedIn Recruiter, made it possible to reach and contact psychiatrists from individual continents faster. The research conducted so far (published in the Journal of Clinical Medicine) has allowed to state that it is a quick form of contact, and thus conducting research. The possibility of a psychiatrist to ask a question or to correspond with him, e.g., regarding the purpose of the examination, is one of the arguments that justified the choice of this form of contact. For the lack of additional portals, see Manuscript limitations.

3. “The term “psychoactive substances” was included since the abstract. However, the survey did not consider any question about these drugs and potential pharmacological interactions with psychotropic medications. Questions were more related to schizophrenia and bipolar affective disorder treatments (assuming as dual diagnoses?).”

ANSWER:This sentence in the abstract has been edited as suggested, i.e., it has been corrected, giving as an example of patients taking new psychoactive substances.

4. „Authors should mention that some differences in therapeutic doses of antipsychotics (e.g. for clozapine in doi: 10.1055/a-1625-6388; DOI: 10.1016/j.euroneuro.2022.03.001) are known to be due to ethnicity (i.e. allele frequencies of pharmacogenes associated to psychotropic medications).”

ANSWER:In line with the suggestion obtained, this aspect and the reference to the indicated publications were included in the discussion.

5. “please, correct some misspellings:

- Line 135 was mainly through the “Lincedin” Recruiter portal.

-The title of all tables appear after the tables are shown.

-In line 450, please, define abbreviations OLZ, and RIS.”

ANSWER:We also thank you for such minor comments. The word LinkedIn was edited with the correct spelling. The table titles are placed above them. The abbreviations indicated have been expanded.

6. „It is urgent to include more discussion of polypharmacy in educational activities for psychiatrists.”

ANSWER:In line with the comments received, the discussion was expanded to include ways to broaden psychiatrist education in terms of drug interactions.

Round 2

Reviewer 1 Report

Manuscript is improved and it should be published in present form.

Reviewer 2 Report

The authors have included gender analyses concerning the psychiatrists sample, however, this does not change the rather descriptive character of the study.